# Neurophysiological Assessment of An Innovative Maritime Safety System in Terms of Ship Operators’ Mental Workload, Stress, and Attention in the Full Mission Bridge Simulator

**DOI:** 10.3390/brainsci13091319

**Published:** 2023-09-14

**Authors:** Vincenzo Ronca, Esma Uflaz, Osman Turan, Hadi Bantan, Scott N. MacKinnon, Andrea Lommi, Simone Pozzi, Rafet Emek Kurt, Ozcan Arslan, Yasin Burak Kurt, Pelin Erdem, Emre Akyuz, Alessia Vozzi, Gianluca Di Flumeri, Pietro Aricò, Andrea Giorgi, Rossella Capotorto, Fabio Babiloni, Gianluca Borghini

**Affiliations:** 1Department of Computer, Control, and Management Engineering, Sapienza University of Rome, 00185 Roma, Italy; vincenzo.ronca@uniroma1.it (V.R.); pietro.arico@uniroma1.it (P.A.); capotorto.1843967@studenti.uniroma1.it (R.C.); 2BrainSigns Srl, Industrial Neurosciences Lab, 00198 Rome, Italy; alessia.vozzi@uniroma1.it (A.V.); gianluca.diflumeri@uniroma1.it (G.D.F.); andrea.giorgi@uniroma1.it (A.G.); fabio.babiloni@uniroma1.it (F.B.); 3Department of Maritime Transportation and Management Engineering, Istanbul Technical University, Tuzla, Istanbul 34485, Turkey; uflaz16@itu.edu.tr (E.U.); arslano@itu.edu.tr (O.A.); eakyuz@itu.edu.tr (E.A.); 4Maritime Human Factors Centre, Histological, Forensic and Orthopaedic Sciences, University of Strathclyde Glasgow, Glasgow G1 1XQ, UK; o.turan@strath.ac.uk (O.T.); hadi.bantan@strath.ac.uk (H.B.); rafet.kurt@strath.ac.uk (R.E.K.); burak.kurt@strath.ac.uk (Y.B.K.); pelin.erdem@strath.ac.uk (P.E.); 5Department of Mechanics and Maritime Sciences, Chalmers University of Technology, 41296 Gothenburg, Sweden; scottm@chalmers.se; 6Cetena SpA, 16121 Rome, Italy; andrea.lommi@cetena.it; 7DeepBlue srl, 00185 Rome, Italy; simone.pozzi@dblue.it; 8Department of Anatomical, Histological, Forensic and Orthopaedic Sciences, Sapienza University of Rome, 00185 Roma, Italy; 9Department of Molecular Medicine, Sapienza University of Rome, 00185 Roma, Italy; 10College of Computer Science and Technology, Hangzhou Dianzi University, Hangzhou 310005, China

**Keywords:** neurophysiology, human factors, mental workload, stress, attention, maritime, safety

## Abstract

The current industrial environment relies heavily on maritime transportation. Despite the continuous technological advances for the development of innovative safety software and hardware systems, there is a consistent gap in the scientific literature regarding the objective evaluation of the performance of maritime operators. The human factor is profoundly affected by changes in human performance or psychological state. The difficulty lies in the fact that the technology, tools, and protocols for investigating human performance are not fully mature or suitable for experimental investigation. The present research aims to integrate these two concepts by (i) objectively characterizing the psychological state of mariners, i.e., mental workload, stress, and attention, through their electroencephalographic (EEG) signal analysis, and (ii) validating an innovative safety framework countermeasure, defined as Human Risk-Informed Design (HURID), through the aforementioned neurophysiological approach. The proposed study involved 26 mariners within a high-fidelity bridge simulator while encountering collision risk in congested waters with and without the HURID. Subjective, behavioral, and neurophysiological data, i.e., EEG, were collected throughout the experimental activities. The results showed that the participants experienced a statistically significant higher mental workload and stress while performing the maritime activities without the HURID, while their attention level was statistically lower compared to the condition in which they performed the experiments with the HURID (all *p* < 0.05). Therefore, the presented study confirmed the effectiveness of the HURID during maritime operations in critical scenarios and led the way to extend the neurophysiological evaluation of the HFs of maritime operators during the performance of critical and/or standard shipboard tasks.

## 1. Introduction

In today’s globalized trade and industrial supply chain, the transport industry, particularly maritime transportation, plays a vital role. The majority of the world’s trade in goods, more than 80%, is carried out using ships [1]. However, this increased reliance on maritime transport has led to a number of issues with the human factors that have a direct impact on maritime safety. The growth in demand for maritime transport has led to the development of larger and faster ships, resulting in a significant increase in maritime traffic and extreme traffic conditions [2]. Despite ongoing efforts by various organizations and authorities to regulate the industry, the inherent complexities and risks associated with maritime transport remain. These risks arise in the form of accidents that can disrupt vital shipping waterways, result in significant financial losses, cause loss of life, and lead to marine pollution [3,4]. Ensuring maritime safety has always been a top priority for the international shipping sector.

Addressing these human factor issues is essential to maintaining a safe and efficient maritime environment. The consequences of failing to manage these factors adequately can be severe, ranging from accidents and environmental damage to detrimental effects on the well-being of crew members. Therefore, understanding and integrating human factor considerations into maritime operations and safety protocols are essential steps to mitigate risk and ensure safety on board.

The annual statistics provided by the European Maritime Safety Agency (EMSA) provide an insight into maritime accidents and incidents. According to their latest report, of the 21,173 accidents and incidents registered between 2011 and 2018, a remarkable 81.1% were attributed to human error [5]. Over the past decade, the maritime industry has experienced a significant number of different emergency situations, including collisions, explosions, fires, flooding, groundings, and man-overboard incidents. Unfortunately, these incidents have resulted in loss of life, loss of goods, and significant environmental damage. Mitigating the impact and consequences of maritime accidents relies heavily on human factors and human response. As global maritime trade expands and the number of vessels navigating congested and narrow waters increases, the likelihood of accidents, particularly collisions, contacts, and groundings, has increased [6,7,8,9].

The contextualization of human factors (HFs) within maritime environments is crucial for understanding their relevance and impact on safety. Human factors encompass various psychological, physiological, and ergonomic aspects that influence human performance in complex work environments [10]. These factors include, but are not limited to, crew performance, communication, decision-making, fatigue, stress, workload management, and situational awareness [11].

In the maritime industry, where safety is of utmost importance, it is essential to explore the role of HFs in accidents occurring in these environments. Specifically, in this case study, we focus on the contextualization of HFs, such as mental workload, stress, and attention by measuring neurophysiological signals [12,13,14,15,16,17,18,19,20,21].

The literature contains numerous studies conducted by researchers examining the influence of organizational and human factors on ship accidents [7,22,23]. These studies have identified various contributing factors to accidents, including fatigue, high workload, stress, physical deprivation, insufficient knowledge, and lack of situational awareness [24]. In emergency situations, non-technical capabilities, including psychological factors, play a significant role in collaborative decision-making processes, which significantly impact maritime safety.

The various psychological states, mental workload, stress, and attention have significant implications for human performance in maritime contexts. Mental workload refers to the cognitive demands imposed on individuals while performing tasks [25]. In the maritime industry, where crews navigate through congested or narrow waterways, maintain situational awareness, and respond to complex scenarios, the mental workload can significantly impact decision-making and performance [24,26,27]. On the other hand, stress can significantly impact human performance, and that can arise from factors such as time pressure, adverse weather conditions, and high-stakes situations [28,29]. The presence of stress can impair cognitive functioning, attentional focus, and response capabilities, thereby increasing the risk of accidents in maritime environments.

Attention is crucial for maintaining situational awareness and detecting critical cues in the maritime setting. Sustained attention and vigilance are necessary to monitor vessel movements, identify potential hazards, and promptly respond to changing circumstances. However, lower attention can occur, leading to critical errors and accidents.

This study focuses on measuring neurophysiological signals to understand better and assess HFs. Neurophysiological signals, such as electroencephalography (EEG), provide a direct and objective assessment of mental states and cognitive processes. By examining the neurophysiological correlates of mental workload, stress, and attention, we gain valuable insights into the underlying mechanisms that influence human performance in maritime environments. In fact, such an approach relying on neurophysiological signals processing is widely considered to be more objective in scientific research with respect to the approaches based on subjective and behavioral measurements. Several previous works already demonstrated how EEG-based indicators are able to objectively characterize the above-mentioned humans’ mental states, i.e., the mental workload, the stress, and the attention. More specifically, refs. [21,30,31,32] demonstrated how humans’ attention and vigilance can be characterized by a synthetic EEG-based index. Regarding the mental workload, several studies [12,33,34,35] showed the reliability of specific EEG-features to the humans’ mental workload while performing different activities in operational environments. Concerning stress evaluation, a plethora of scientific research revealed the sensibility of EEG correlates to such variations in mental state while performing different operational tasks in both real and realistic environments [36,37].

In this case study, we aim to contextualize the impact of a human factors analysis approach (i.e., HURID), specifically mental workload, stress, and attention, on accidents occurring in maritime environments. Through the assessment of neurophysiological signals, we strive to deepen our understanding of how these factors influence human performance and contribute to maritime accidents. By identifying the key HFs and their impact, we can develop targeted interventions and strategies to enhance safety, reduce the risk of accidents, and promote efficient operations in maritime settings. The present paper introduces an innovative approach, the assessment of neurophysiological signals, which represents the first application of its kind in the maritime domain regarding collisions in congested waters with a detailed experimental approach. The approach has been tested to demonstrate its effectiveness in the context of handling ship collision risk in congested water.

### 1.1. HURID (Human Risk-Informed Design)

There are some challenges to assessing the potential impact of new technologies and human performance in safety, due to the scarcity of human factors data obtained from the investigation of safety events (accident/incident/near miss) and the lack of effective feedback loops from operations back to designers. HURID (Human Risk-Informed Design) is a bridge facilitating the integration of human factors into the design of systems and operations, safety assessment, and regulation [38]. The main purpose of HURID is to serve as a collection of various components, specifically the SHIELD incident database [39], the Human Factors Toolkit, and the Risk Models. The implementation of these components follows a structured procedure that aligns with conventional product development methodologies, encompassing stages such as development, prototyping, and testing prior to production or operational implementation. The utilization of the HURID Framework within the aviation and maritime domains was demonstrated by conducting targeted case studies within these specific domains in the simulator. The primary objective of these case studies was to validate the efficacy of HURID interventions concerning the particular scenarios being examined. Within this framework, the significance of the human factor and human responses were explored in emergency scenarios by employing real-time, full-mission bridge simulator. Extensive investigations have been conducted to study human behaviors and response actions from various perspectives in the context of emergency situations by using subjective and objective methods. Throughout the experimental investigations, the cognitive workload, stress levels, and attention were observed by employing EEG within a bridge simulator. Simulated operational tasks were employed to analyze and gain insights into human behavior under normal conditions as well as during emergency situations, both with and without the implementation of HURID intervention. Furthermore, potential outcomes resulting from these analyses were revealed.

### 1.2. Validation of the HURID

A Lookout is a person assigned to the bridge of a ship with the primary duty of keeping constant watch over the surrounding sea in order to immediately notify the navigation staff of any possible threats that might endanger safe navigation or the ship. A Lookout attends training to learn the core watchkeeping information and abilities before taking on the duty of a Lookout. A Lookout is required by the COLREG (International Regulations for Preventing Collisions at Sea) to always give the ship’s navigation their full attention, making sure that any observed entities—such as other vessels, fishing boats, navigational aids, or floating objects—are promptly reported to the officer on watch (OOW) [40].

The Lookout’s presence on the ship’s bridge and their presentation of thorough information about the surrounding area greatly assist the OOW in doing their tasks successfully [41]. Furthermore, in order to retain maximum attention, the person given watch duty must avoid engaging in any other activities. Maintaining a good Lookout throughout the watch is considered part of the considered work on board a vessel as per COLREG Rule-5 and Part 3-1 of STCW95 [40,42]. The officer on watch (OOW) is tasked with safely operating the vessel while also performing the Lookout duties during daylight hours. However, it is necessary to have a dedicated Lookout on duty when there is darkness or poor vision. The Master of the vessel may designate an assistant as an extra Lookout depending on the navigating conditions, such as congested waters or heavy weather conditions.

Regarding the function of the only Lookout, the related responsibilities, and degree of watchkeeping, many rules, suggestions, and standards have been created to guarantee safe navigation. However, as noted in a study by the Marine Accident Investigation Branch (MAIB) in 2004 [43], there is a gap between the actual work performed and the specified tasks connected to the sole Lookout and its responsibilities.

The watchkeeping officer must be accompanied by a committed Lookout to maintain efficient watchkeeping. The Lookout must refrain from taking on any additional obligations and concentrate entirely on the task they have been given. Such support for the principal operator should also remain around the specified duty area until their shift is over. In congested waterways with high traffic density, such as the Dover Strait, Gibraltar Strait, and Singapore Strait, the presence of a Lookout on the bridge is essential. In these situations, the Lookout plays a crucial role as a bridge team member, reporting to the officer on watch (OOW) on recognized targets and possible dangers and allowing the preservation of situational awareness about the potential threats in the surroundings. The “sole Lookout” factor was the most commonly recognized contributing factor in an inquiry involving 33 incidents and 41 boats between the years 1994 and 2003, according to the Marine Accident Investigation Branch’s (MAIB) research on bridge watchkeeping safety [43]. This report emphasizes the significance of having a Lookout; in other words, adding an extra person onboard for specific conditions on the bridge.

### 1.3. Scenario Selection

According to a report by EMSA 2022 [1], between 2011 and 2022, collisions accounted for the largest proportion of accidents investigated, about 44% of all accidents, followed by groundings (38%) and contacts (18%). Ship collisions are the most common type of maritime accident, resulting in fatalities, injuries, and damage to ships and the environment. Human error was found to be a contributing factor in approximately 78% of collisions, groundings, and collisions investigated.

On the other hand, it is of paramount importance to observe and identify the mental and physical state of the seafarers. The selection and design of the most appropriate scenario is crucial when conducting experimental studies. Therefore, a collision accident in the Strait of Dover, widely recognized as one of the busiest waterways in the world, was selected from the accident database. The experiments were designed not only to observe human performance during the experiments in the full mission ship bridge simulator, but also to measure the mental state of the participants. The mental state of the participants was monitored using an EEG device within the bridge simulator. The same collision scenario involving the same ship was used for all participants, requiring them to perform a series of actions.

### 1.4. Background of An Accident Selected for Analysis Using HURID

In October 2008, a collision took place in the Dover Strait involving the bulk carrier MV Wadi Halfa and the general cargo vessel MV Scott Isles at 4:49 a.m. Figure 1 displays the collision approach on the map obtained from the accident report. The incident occurred under good visibility and calm weather conditions, with congestion in the navigational area contributing to increased traffic. As outlined in the accident report by the MAIB, 2009 [44], the processes leading to the accident can be summarized as follows using the critical incident technique.

While crossing the NE traffic lane of the Dover Strait Traffic Separation Scheme during its passage from Rochester to Antwerp, the watchkeeping officer on MV Scot Isles failed to detect the approaching vessel MV Wadi Halfa prior to the collision. Despite attempting evasive maneuvers, the collision could not be avoided. Notably, there were no Lookouts on either bridge during the collision, and the radar and other bridge equipment on both vessels were not utilized effectively. The accident analysis of the collision, conducted using critical incident techniques, identified several critical aspects that contributed to the incident. One of the most crucial findings was the absence of a Lookout on both vessels’ bridges at the time of the collision. The lack of dedicated personnel keeping a vigilant watch compromised the ability to detect and respond to potential collision risks. Additionally, it was highlighted that the radar systems and other bridge equipment on both vessels were not effectively utilized, further limiting situational awareness and early detection of the approaching vessel. These factors combined to create a significant gap in the vessels’ ability to prevent the collision. The analysis underlines the importance of maintaining proper Lookout practices and utilizing navigational equipment to ensure the safety of vessels navigating in congested waters like the Dover Strait.

### 1.5. Research Questions

This study generated a set of research questions (RQs) that will provide the foundation for examining the effects of how the HURID approach might improve the safety of navigation. The following questions will be assessed:

RQ 1: What are the differences in reaction times experienced by participants when they encounter collision risks in congested waters between two conditions—sole Lookout and with Lookout?

RQ 2: How does operational safety, specifically the number of collisions, differ between two conditions—sole Lookout and with Lookout—when participants encounter collision risks in congested waters?

RQ3: What are the differences in terms of mental workload experienced by participants under collision risks in congested waters between two conditions—sole Lookout and with Lookout?

RQ 4: How do stress levels vary among participants when they encounter collision risks in congested waters between two conditions—sole Lookout and with Lookout?

RQ 5: What variations occur in participants’ attention levels under collision risks in congested waters between two conditions—sole Lookout and with Lookout?

## 2. Materials and Methods

This section is constituted by different subparagraphs in order to properly describe the experimental scenario design, the sample size involved in the experimental activities, and the data collection procedures.

### 2.1. Experimental Scenario Design

The past-accident case study was run in a real-time, full-mission ship bridge simulator to explore the significance of human factors and examine human responses in emergency contexts. The study focused explicitly on ship collision scenarios. Throughout the experimental investigations, the designated individual’s cognitive workload, stress levels, and attentional states were meticulously monitored using EEG within the bridge simulator environment. Simulated operational activities were deployed to comprehensively analyze and gain insights into human behavior in both baseline and emergency situations, with and without the implementation of HURID intervention. The experiment involved two stages of simulator tests aimed at evaluating the effectiveness of HURID. Specifically, the intervention chosen for verification was the increase in manning level by introducing a Lookout to support OOW. Introducing the Lookout is not only in line with the company’s ISM procedures but also aligned with the findings of the SHIELD and Collision Risk Model analyses. Therefore, the proposed intervention was deemed most suitable for the scenario.

In the initial phase of the experiment, 13 participants were placed in the Dover Strait simulation without the implementation of the HURID intervention. Throughout the simulation, the participants were tasked with managing the situation without a bridge Lookout, testing their ability to handle the scenario independently.

In the second iteration of the experiment, the simulator was operated with the HURID intervention, which involved increasing the manning level. The experiments involved 13 new participants, with the same grade of skills and experience in the maritime context, while an additional crew member, i.e., the Lookout, with maritime experience was the Lookout in all scenarios. The assigned Lookout was the same for every experiment, and the Lookout had the specific task of timely reporting during the simulation, ensuring that all targets were reported consistently across all sessions.

This decision aimed to replicate the conditions found in the accident report, where both vessels had a Lookout during the night watch. However, prior to the accident, both officers granted permission to their respective Lookouts to leave the bridge temporarily for various reasons, such as cleaning, checking provisions, supporting technical personnel, or safety rounds. The full-mission ship bridge simulator laboratory at the University of Strathclyde (UoS) served as the setting for the creation of the scenario, which was designed and developed based on the accident report. A general view of the simulator experiment is shown below in Figure 2.

The schematic representation of the scenario is illustrated in Figure 3. The main vessel deployed for the purpose of role-sharing among participants was identified as MV Wadi Halfa. A pre-defined speed and course were assigned to MV Scott Isles, designating it as the target vessel. To ensure the fidelity of the navigational environment within the Dover Strait, the scenario design was meticulously aligned with the comprehensive vessel details outlined in the accident report.

### 2.2. Participants

A total of 26 participants were engaged in this case study to ascertain the efficacy and effectiveness of the HURID intervention. By examining the performance of individuals in various simulated emergency situations, the study aimed to evaluate the impact and efficacy of the HURID intervention on human responses in ship collision scenarios. Each participant (OOW) was fitted with the EEG to ensure compliance with data protection regulations, operators provided comprehensive information regarding the experiment and demonstrated adherence to the General Data Protection Regulation (EU regulation 2016/679). Before the experiments, the participants were thoroughly briefed on utilizing their personal information, after which they willingly provided informed consent by signing consent forms and an orientation script. An additional interactive familiarization process was implemented to familiarize the participants with the specific scenario, replicating the same ship and environmental conditions as the actual scenario. A demographic questionnaire was administered to gather general participant information. Once participants indicated their readiness, they were equipped with an EEG device for calibration. In the second phase of the simulator study, participants executed the designated scenario with a Lookout. Following the conclusion of the scenarios, participants engaged in open-ended discussions during the debriefing session regarding their experiences with the simulator trials and the maneuvers executed. The NASA Task Load Index (NASA-TLX) [45] was also employed as a subjective measure to assess the perceived workload. Furthermore, participants were requested to complete a post-scenario questionnaire as part of the debriefing process.

### 2.3. Experimental Data Collection

During the validation experiments conducted, both subjective and objective measures were implemented to assess the effectiveness of the HURID. The first experiment involved a total of fifteen participants, while the second experiment included nineteen participants. However, due to signal artefacts and technical issues, certain participants had to be excluded. We finally obtained thirteen participants for the “no HURID” and thirteen participants for the “HURID” experimental condition.

#### 2.3.1. Subjective and Behavioural Data Collection and Analysis

The experimental protocol foresaw subjective and behavioral data collection. In particular, the number of Very High Frequency (VHF) radio calls made by the participants while performing the maritime tasks was collected. The number of VHF calls made by the participant indicates the extent to which the OOW tried to contact other vessels and shore support to handle potential risky situations arising during maritime activities. Moreover, the reaction times exhibited by the participants to properly respond to the different events during the maritime simulation were collected. In this context, the number of collisions experienced by participants during the experimental activities was also recorded. Finally, the NASA-TLX questionnaire was performed by each participant at the end of the experimental session. The NASA-TLX is a widely recognized and extensively used measurement tool that evaluates the subjective workload experienced by individuals performing complex tasks, especially in high-pressure environments. It provides valuable insights into the mental and physical demands placed on individuals during task execution, allowing for the identification of potential areas of improvement and optimization in task design, workload distribution, and resource allocation [46]. The above-described NASA-TLX, reaction times, number of VHF calls, and number of collisions were included in the statistical analysis. Before performing the analysis, the number of calls made, and the number of collisions were combined in a unique performance index as follows:$$Performance\;index=1-(\frac{CallNum-min_{CallNum}}{max_{CallNum}-min_{CallNum}}+\frac{CollNum-min_{CollNum}}{max_{CollNum}-min_{CollNum}})\cdot 0.5$$
where *CallNum* represents the number of VHF calls, and *CollNum* represents the number of collisions. The presented index was defined by assigning the same significance to the factor related to the VHF calls and the one related to the collisions number [47].

#### 2.3.2. Neurophysiological Signal Collection and Analysis

The ship operators’ EEG signals were recorded by the digital monitoring system LiveAmp (Brain Products, Germany) with a sampling frequency of 125 (Hz). The eight water-based recording electrodes were properly placed over the frontal and parietal brain areas commonly considered for mental state assessment [30,35,48,49]. In particular, the EEG channels were the following ones: AFz, AF3, AF4, AF7, AF8, Pz, P3, P4, all referenced to the left mastoid and grounded to the right mastoid. Once the electrodes’ impedances (kept below 50 (kΩ)) and the quality of the EEG signals were checked, the experimental protocol started.

The EEG signal was first band-pass filtered with a 5th-order Butterworth filter in the interval 2–30 (Hz). The eye blink artefacts were detected and corrected online by a modified implementation of the Multi-Channel Wiener filtering through the Reblinca method [50]. For further sources of artefacts, specific algorithms of the EEGLAB toolbox [51] were applied. Specifically, the preprocessed EEG signal has been divided into 1 s long epochs. Three criteria have been applied to recognize artifactual data automatically. Firstly, EEG epochs with the signal amplitude exceeding ±80 μV (Threshold criterion) were marked as “artefacts’’. Then, each EEG epoch was interpolated to check the trend’s slope within the considered epoch (Trend estimation). If such a slope is higher than 20 μV/s, the considered epoch is marked as “artefact”. Finally, the signal sample-to-sample difference (Sample-to-sample criterion) was analyzed: if such a difference, in terms of absolute amplitude, was higher than 25 μV, i.e., an abrupt variation (no-physiological) happened, the EEG epoch was marked as “artefact”. In the end, the EEG epochs marked as “artefacts” were removed from the EEG dataset with the aim of having a clean EEG signal to perform the analyses.

From the artefact-free EEG, the Global Field Power was calculated for the EEG frequency band of interest for the mental state evaluation, which was the Theta, Alpha, and Beta. The GFP was chosen as the parameter of interest describing brain EEG activity since it has the advantage of representing, in the time domain, the degree of synchronization or a specific cortical region of interest in a specific frequency band [52,53,54]. More specifically, the GFP was mathematically computed according to the same approach described by Vecchiato and colleagues [55]. The EEG frequency bands were defined according to the Individual Alpha Frequency (IAF) value [56] computed for each participant. Since the Alpha peak is mainly prominent during rest conditions, the subjects were asked to keep their eyes open for one minute before starting the experiment. Such a condition was then used to estimate the IAF value specifically for each participant. The GFP was calculated over all the EEG channels for each epoch using a Hanning window of the same length of the considered epoch (1 s, which means 1 Hz of frequency resolution according to the time resolution required from the presented approach). After the EEG data preprocessing, the EEG GFP-derived features were computed to objectively characterize the relevant mental states within the above-described experimental protocol design. In particular, the mental workload, the stress, and the attention indexes were computed as follows:$$Mental\;workload=\frac{Frontal\;Theta_{GFP}\{Af3,\;Afz,\;Af4\}}{Parietal\;Alpha_{GFP}\{P3,\;Pz,\;P4\}}$$
$$Stress=Parietal\;Beta\;High_{GFP}\{P3,\;P4\}$$
$$Attention=\frac{Frontal\;Beta_{GFP}\{Af8,\;Af4,\;Afz,\;Af3,\;Af7\}}{Frontal\;Theta_{GFP}\{Af8,\;Af4,\;Afz,\;Af3,\;Af7\}}$$

In this regard, it has to be noted that computation of the mental workload and stress indexes were defined according to different studies carried out previously in which such mental states were deeply investigated as EEG-derived features [12,57,58]. Similarly, the attention index definition was selected according to the inverse of the so-called Theta-Beta Ratio, an EEG-derived feature broadly validated as an indicator of ADHD [59,60,61].

#### 2.3.3. Statistical Analysis

The statistical analysis was performed after the data normalization. More specifically, the subjective and behavioral data, i.e., the NASA-TLX, reaction times, and the number of VHF calls, were normalized according to their respective maximum and minimum values throughout the entire experimental session. The neurophysiological derived features, i.e., mental workload, stress, and attention, were normalized through the z-score technique according to the entire experimental session as follows:$$X_{z-scored}=\frac{X_{i}-median(X)}{mad(X)}$$
where *X* corresponds to the neurometrics distributions and *mad* corresponds to the median absolute deviation.

Subsequently, the Shapiro–Wilk test was used to assess the normality of the distribution related to each of the considered parameters. If normality was confirmed, Student’s t-test was performed for the comparison of independent conditions (i.e., no HURID vs. HURID). In the case of non-normal distribution, the Mann–Whitney test was performed.

## 3. Results

This section is divided into subheadings to provide a concise and precise description of the experimental results in terms of subjective, behavioral, and neurophysiological data.

### 3.1. Subjective Results

The statistical analysis performed on the NASA-TLX revealed no statistical differences (*p* = 0.52) between the no-HURID and the HURID groups. In other words, no statistical differences were observed, in terms of perceived mental workload demand, between the participants who experienced the HURID (with Lookout) and no HURID, those who performed the maritime operations as sole Lookout. The results indicate that this method might not be effective for simulator studies involving different groups of participants. Furthermore, in the experiments conducted with and without HURID, the involvement of different individuals had an impact on the results in terms of subjectivity.

### 3.2. Behavioral Results

The independent sample t-test performed on the reaction times revealed that the participants who performed the maritime operations with the HURID (with Lookout) were significantly faster than the ones who performed the operations as sole Lookout. In particular, the reaction times associated with the HURID group were statistically significantly lower than the ones associated with the no-HURID group (*p* = 0.01) (Figure 4). In this regard, it has to be highlighted that the operators included in the HURID group reacted 22.5% faster than the ones included in the no-HURID group.

These findings confirm the role of a Lookout in providing assistance to the officer on watch. Specifically, the presence of a Lookout enables earlier reactions concerning collision prevention action.

Similarly, Figure 5 reveals that the performance index associated with the HURID group resulted in being significantly higher than the one associated with the no-HURID group (*p* = 0.007). Regarding the performance index improvement observed for the HURID group, the overall performance increase was 74.6% for the operators who conducted the maritime simulation tasks with the support of the Lookout compared to the no-HURID group. The collision rate is the most crucial performance index in both groups, and the presence of a Lookout was found to result in a lower collision rate. This indicates that having a Lookout on the bridge has a positive impact on reducing the likelihood of collisions and enhancing overall safety.

### 3.3. Neurophysiological Results

The following Figure 6, Figure 7 and Figure 8 represent the statistical analysis results related to the comparison between the two experimental conditions, e.g., HURID and no HURID, in terms of ship operators’ mental workload, stress, and attention. In particular, the Mann–Whitney test showed a statistically significant increase in the ship operators’ mental workload during the no-HURID experiments compared to the HURID ones (*p* = 0.03). This outcome indicates that the presence of the Lookout consistently reduced the operators’ mental workload while dealing with the maritime operations included within the experimental protocol (Figure 6).

Similarly, the Mann–Whitney test for the independent sample revealed that the operators’ stress consistently and significantly increased while performing the experimental tasks without the HURID compared to the condition in which they were supported by the Lookout (*p* = 0.008). Such a stress increase resulted to be even more consistent with respect to the mental workload, and it indicates that the absence of the Lookout played a crucial role in stressing the operators while performing the maritime operations (Figure 7) in emergency situations.

The statistical analysis revealed a statistically significant decrease in the ship operators’ attention within the no-HURID condition compared to the one with HURID (*p* = 0.01) (Figure 8). This result also confirms the findings of the previous EEG-based parameters that people in HURID condition demonstrated better attention performance.

## 4. Discussion

The collision rate serves as the primary criterion utilized for the validation of the HURID system. Both scenarios, with and without HURID validation, were studied and compared regarding their collision rates. The experimental investigation involved a total of 26 participants. The influence of the lookout in terms of enhancing bridge manning is illustrated in Figure 9.

Taking early action is crucial to prevent collision risks. Therefore, in the HURID scenario, the presence of a Lookout aided the participant in taking prompt and appropriate action (RQ 1) (Figure 4).

The implementation of HURID intervention, specifically increasing bridge manning, resulted in a 61.5% reduction in collision probability (RQ 2). This outcome demonstrates that the HURID intervention significantly mitigates collision risks by enhancing bridge manning. In integrated bridge systems, numerous parameters are available to assess the surrounding environment. However, it becomes challenging for the Officer On Watch (OOW) to accurately evaluate the situation when operating alone in congested or narrow waters with high traffic (Figure 5).

The simulator-based evaluation aimed to test HURID-based interventions aimed at improving the response of the onboard crew during emergencies. The real accident, which simulated the Dover Strait collision accident, was designed and executed in a full-mission bridge simulator equipped with EEG equipment. The objective measures, i.e., the EEG, provided statistical evidence supporting the effectiveness of increasing the manning level on the bridge in enabling the OOW to take appropriate actions early and prevent collisions. In fact, the experimental results revealed that the operators’ mental workload and stress were significantly higher when performing the maritime operations without the Lookout (i.e., the noHURID condition) (Figure 6 and Figure 7) (RQ 3). This indicates that the presence of the Lookout (i.e., the HURID condition) consistently reduced the impact of the critical maritime operations on the operators’ mental states. In other words, the results indicated that the presence of the Lookout can play a crucial role in terms of crew’s safety and efficiency by reducing the mental workload and stress experienced by the main actor of the vessel’s operations while dealing in critical and non-critical scenarios (RQ4). This observation is also supported by the results related to the EEG-based attention index computed along the noHURID and HURID conditions (Figure 8). The maritime operators were objectively and significantly more attentive toward the experimental scenario when they were supported by the Lookout (RQ5).

The HURID intervention resulted in reduced workload and stress, as objectively measured. The Lookout’s reporting further facilitated the OOW’s understanding and assessment of the situation during the experiments.

It has to be observed that despite the positive and promising results the presented work was characterized also by some limitations. In particular, the sample size appeared to be not large. In this regard, it has to be noted that it results to be challenging to recruit such a specific target of participants to be involved in a long and complex experimental protocol in a simulated environment. Furthermore, the presented evaluations were computed overall the experimental conditions. An important next step of the proposed methodology will increase the time resolution of the mental states while performing maritime operations in critical environments.

However, key findings from the simulator experiments include the possibility of objectively measuring various parameters during watchkeeping, such as workload, stress, and visual attention through the use of neurophysiological-based indicators such as the EEG-based parameters. Excessive mental workload and stress can impair susceptibility, delay reaction times, and increase the likelihood of wrong decisions and actions during emergencies. Results from the experiments demonstrated that the Lookout intervention improves the performance of OOW significantly and assists the OOW in taking early actions to prevent collisions. Moreover, the OOW had more time to utilize the electronic equipment and assess the situation with reduced mental workload and stress due to the HURID intervention.

## 5. Conclusions

Human error is the most prevalent cause of maritime accidents, consequently, the human factor is the primary focus of maritime safety. The Lookout is an integral and crucial component of the bridge team. The meaning of “Lookout” appears traditional or possibly useless. On the contrary, the ship’s “Lookout” role remains highly important, particularly in congested waterways, narrow water, extreme conditions, and heavy traffic.

Collisions and grounding incidents still occur despite the complex bridge system and cutting-edge technology. Human error contributes to the great majority of accidents. Workload, stress, and lack of attention can all contribute to human error. During an emergency, the OOW needs to track the number of parameters on the displays in order to take immediate action to prevent a collision or dangerous situation.

The aim of the study was to investigate the significance of Human Factors and examine human responses in emergency scenarios within the maritime context by using real-time, full-mission bridge simulators. Specifically, simulations were conducted to analyze and gain insights into human behavior under normal and emergency conditions, taking into account the presence or absence of HURID interventions in procedural design. In this regard, we designed ship collision situations and employed a bridge simulator equipped with EEG technology to assess the mental workload, stress, and attention of participants with and without the HURID.

The findings derived from the simulator experiments hold substantial value for maritime stakeholders and regulatory bodies, as they can inform the revision of international rules and regulations. Furthermore, the outcomes can prompt shipping companies to implement the measures in terms of bridge manning levels within their Safety Management Systems properly and according to their ISM procedures, particularly in critical areas. Similarly, the proposed approach could lead to a deeper investigation of the most critical maritime operations in terms of required mental demands and, therefore, focusing the training on such directions. In this regard, it has to be highlighted that the training optimization would have a relevant benefit, especially in terms of safety but also in terms of overall economic cost associated with the training programs. It may be emphasized that implementing ISM procedures appropriately will improve the crew’s performance and enhance navigational safety. These results highlight the significance of adhering to ISM protocols, which can lead to a more competent and efficient crew, ultimately contributing to a safer maritime environment.

## Figures and Tables

**Figure 1 brainsci-13-01319-f001:**
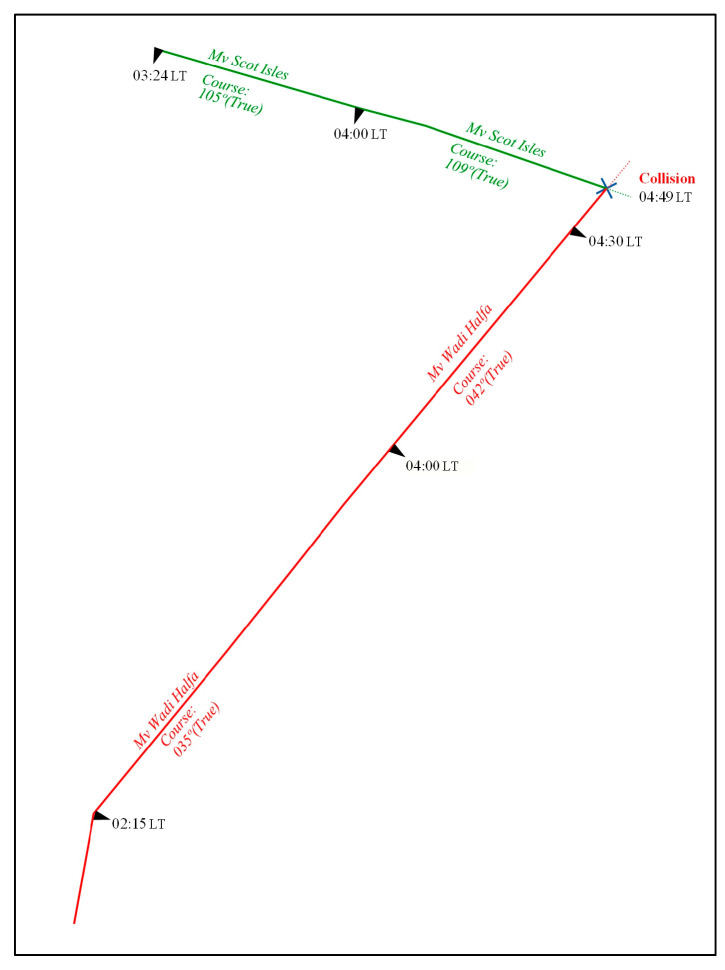
MV Wadi Halfa and MV Scot Isles collision trajectories over time (Retrieved from MAIB) [44].

**Figure 2 brainsci-13-01319-f002:**
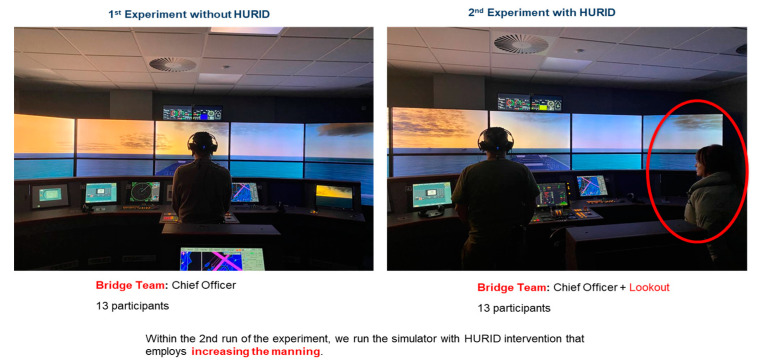
Bridge view without HURID and with HURID. In the latter condition, the Lookout (highlighted in the red circle) was standing on the right side of the Chief Officer.

**Figure 3 brainsci-13-01319-f003:**
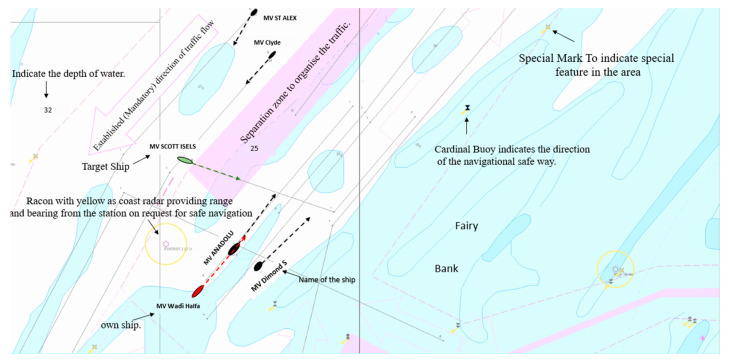
Navigational environment for the selected Dover Strait scenario.

**Figure 4 brainsci-13-01319-f004:**
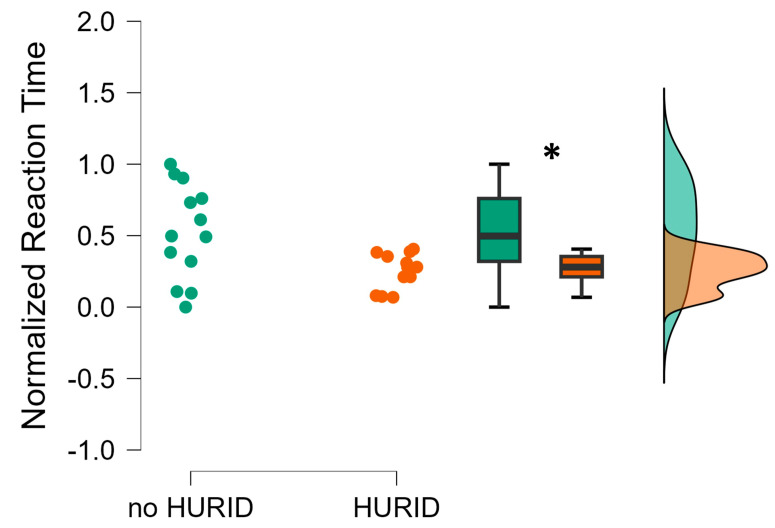
The normalized reaction time exhibited by the operators during the maritime tasks execution without (noHURID) and with (HURID) the Lookout. It is possible to see how the averaged Chief Officers’ reaction in the HURID condition (orange color) was significantly faster than in the noHURID (green color). The asterisk shows that the statistics reported a *p* < 0.05.

**Figure 5 brainsci-13-01319-f005:**
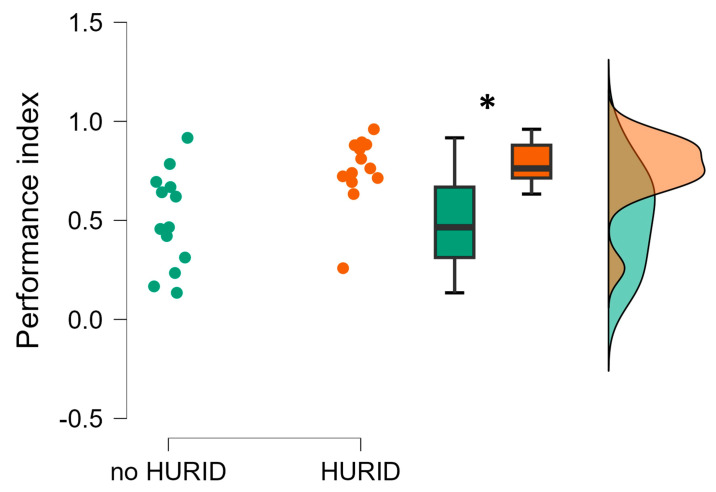
Performance index characterizing the operators’ maritime tasks execution without (noHURID) and with (HURID) the Lookout. It is possible to see how the averaged Chief Officers’ performance in the HURID condition (orange color) was significantly higher than in the noHURID (green color). The asterisk shows that the statistics reported a *p* < 0.05.

**Figure 6 brainsci-13-01319-f006:**
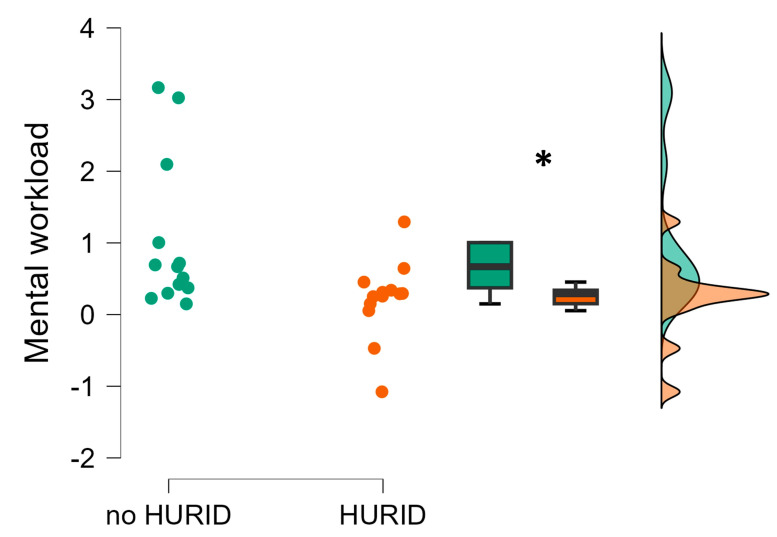
Operators’ EEG-based mental workload index evaluated during the maritime tasks performed without (noHURID) and with (HURID) the Lookout. It is possible to see how the averaged Chief Officers’ mental workload in the HURID condition (orange color) was significantly lower than in the noHURID (green color). The asterisk shows that the statistics reported a *p* < 0.05.

**Figure 7 brainsci-13-01319-f007:**
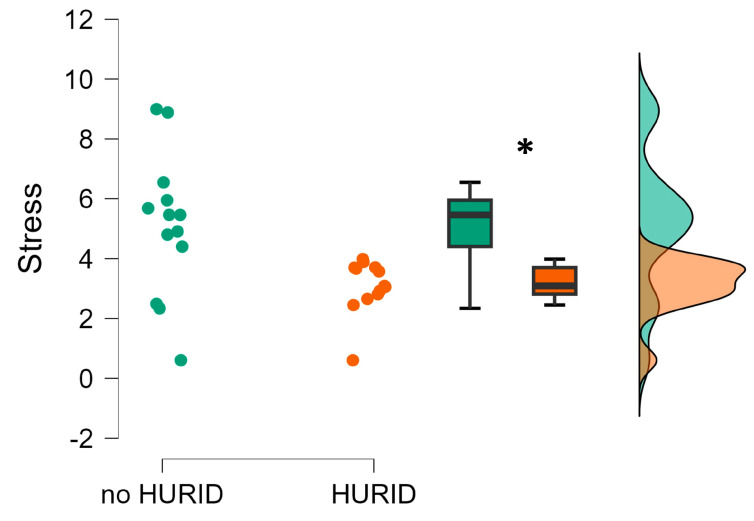
Operators’ EEG-based stress index evaluated along the conditions in which they performed the experiments without (noHURID) and with (HURID) the Lookout. It is possible to see how the averaged Chief Officers’ stress in the HURID condition (orange color) was significantly lower than in the noHURID (green color). The asterisk shows that the statistics reported a *p* < 0.05.

**Figure 8 brainsci-13-01319-f008:**
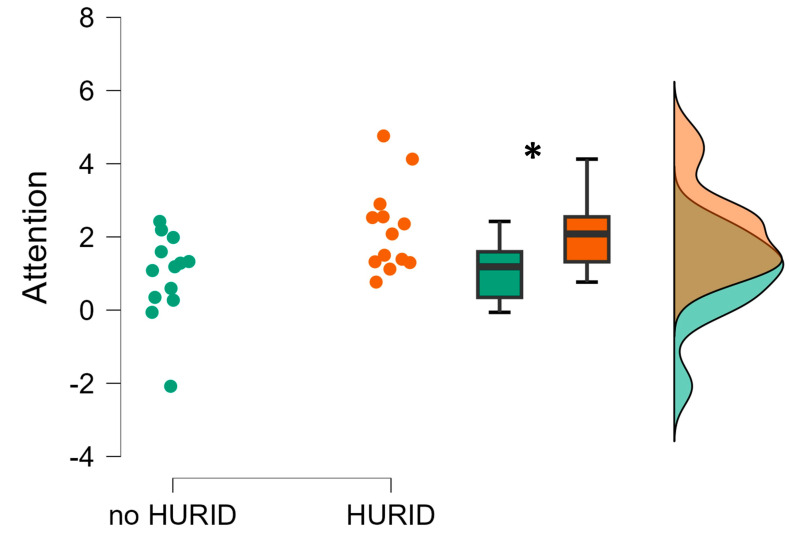
Operators’ EEG-based attention index evaluated along the conditions in which they performed the experiments without (noHURID) and with (HURID) the Lookout. It is possible to see how the averaged Chief Officers’ attention in the HURID condition (orange color) was significantly higher than in the noHURID (green color). The asterisk shows that the statistics reported a *p* < 0.05.

**Figure 9 brainsci-13-01319-f009:**
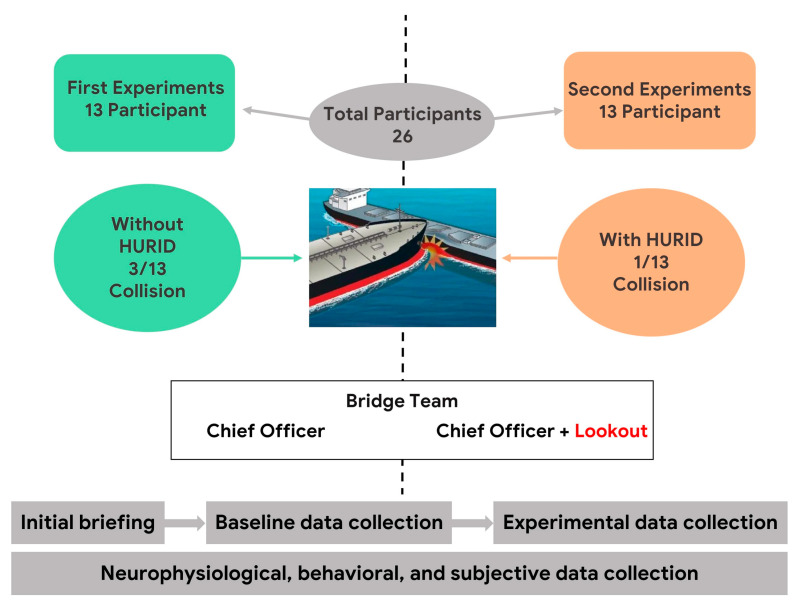
HURID validation behavioral parameters and experimental protocol workflow.

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
