# Peer review of "Neurophysiological Assessment of An Innovative Maritime Safety System in Terms of Ship Operators’ Mental Workload, Stress, and Attention in the Full Mission Bridge Simulator"

_brainsci, 2023, doi:10.3390/brainsci13091319_

Round 1

Reviewer 1 Report

This work aims to assess the psychological state of mariners through EEG signal analysis and validate a framework countermeasure named HURID. The results showed that the participants experienced a statistically significant higher mental workload and stress while performing the maritime activities without the HURID, while their attention level was statistically lower compared to the condition in which they performed the experiments with the HURID. The motivation and objective of this work are clear and interesting, and the proposed technique sounds available. However, several vital issues should be further clarified and improved before acceptance, as listed below:      

1. The manuscript exhibits lots of acronyms, which pose challenges for readers in terms of fluency and comprehension. Therefore, it is desired to include a table that clarifies the acronyms employed, provides their corresponding full names, and offers a succinct description of their fundamental usage. The incorporation of such a table would significantly enhance the convenience and accessibility of the manuscript. 

2. Please label the equation used in this work. As for the performance index, what is its mathematical meaning? Why 0.5 is needed here? Further explanation of such an index is needed.

3. Concerning EEG signal processing, generally, there are five brain rhythms in EEG-based studies. Why only use three of them (Theta, Alpha, and Beta) in this work? Besides, what are the frequency ranges of them? Moreover, why employ a Hanning window with a 1-second length? Rather than other window functions and time lengths?

4. There are several channels in one region, such as five channels (AFz, AF3, AF4, AF7, AF8) in the Frontal region, then how to obtain a GFP value from either the Frontal or Parietal region? Please provide the equations to calculate all elements in the indexes of mental workload, stress, and attention, respectively. For example, how to calculate Frontal Theta GFP, Parietal Alpha GFP, etc. I cannot find any information about them mathematically.  

5. What if the applied number of channels increases or decreases? Have more obvious results been obtained or not? Is the EEG channel number influence the statistical analysis results? Why?

6. To illustrate the experiment in this work, the authors should draw a brief workflow. The current presentation is not easy to follow.

7. From Figure 4 to Figure 8, it seems to me that several results from the two groups overlap in a very close range. Have the authors performed ANOVA tests to validate their findings? What are the ROC curves and AUC results? How to avoid individual differences in EEG-based studies?    

8. To facilitate reproducible research, I strongly recommend that the authors release their EEG data on github.com or other websites that can be freely assessed. It would make a positive effect on the academic community.

9. Regarding future work, could the authors describe it in the paper? What are the usages of such findings? Is it possible to design a classification or recognition tool based on the current findings in this work?

The authors should spend time revising this manuscript, the technical writing and the flow of this work needs to be further improved.

Reviewer 2 Report

This is a well-structured article. The main question addressed by this research is the neurophysiological assessment of an innovative maritime safety system regarding ship operators' mental Workload, stress and Attention in the full mission bridge simulator.

The introduction gives the background of this study as it briefly describes maritime transportation, human and organizational factors affecting safety and relevant literature data as well as the main research questions of the present study.

“Materials and Methods” section is descriptive enough. It refers to the experimental scenario design, the participants and the procedure of data collection.

The results are quite interesting and, to my opinion, well presented and depicted in related figures.

The discussion is well written, summarizing and discussing the main findings of the study and trying to correlate it with recent relative studies. I think that a paragraph summarizing the main limitations of the study would, to my opinion, add to the scientific value of this paper.

Regarding “conclusions”, a section proposing some specific targets for future studies could be added.

References, although relatively few, are relative to the subject.

English language and style are generally fine but there are some minor issues that need to be addressed before publication (for example some long sentences could be separated in shorter ones, to make the text more comprehensible).

Reviewer 3 Report

This study investigated the feasibility of Human Risk-Informed Design (HURID) in the ship operations. The authors calculated the performance index by using NASA Task Load Index, Very High Frequency radio calls and number of collisions. They also estimated the mental workload, stress and attention by usin EEG. The results showed a significant difference in these indicators between two subjects groups with and without HURID, each of which consisted of 13 subjects.

This is an interesting study and might be useful  for considering safer ship operations. Here are my comments;

1) It would be informative to explain how the EEG channel selected, e.g., why AF electrodes instead of F electrodes, which are frequently used in the international 10-20 method. Also the authors used different numbers of channels for frontal and parietal 5 and 3).

2) It would be of interest how the authors analyzed EEG data in more detail. How many epochs of 1s were used? How the signals from multiple channels were treated for calculating the global field power?

3) Since the number of subjects is relatively small, the skills, experience and personality may be an important factor in their decision and behavior. It would be useful to show there was no difference between the two groups in these points.

Round 2

Reviewer 1 Report

The authors have completed the revisions, and the technical content is good. This manuscript is clearly written and justified, so it can be accepted.